# *MnHR4* Functions during Molting of *Macrobrachium nipponense* by Regulating 20E Synthesis and Mediating 20E Signaling

**DOI:** 10.3390/ijms232012528

**Published:** 2022-10-19

**Authors:** Huwei Yuan, Wenyi Zhang, Hui Qiao, Shubo Jin, Sufei Jiang, Yiwei Xiong, Yongsheng Gong, Hongtuo Fu

**Affiliations:** 1Wuxi Fisheries College, Nanjing Agricultural University, Wuxi 214081, China; 2Key Laboratory of Freshwater Fisheries and Germplasm Resources Utilization, Ministry of Agriculture and Rural Affairs, Freshwater Fisheries Research Center, Chinese Academy of Fishery Sciences, Wuxi 214081, China

**Keywords:** *MnHR4*, 20-hydroxyecdysone (20E), *Macrobrachium nipponense*, molt, RNA interference

## Abstract

*HR4*, a member of the nuclear receptor family, has been extensively studied in insect molting and development, but reports on crustaceans are still lacking. In the current study, the *MnHR4* gene was identified in *Macrobrachium nipponense*. To further improve the molting molecular mechanism of *M. nipponense*, this study investigated whether *MnHR4* functions during the molting process of *M. nipponense*. The domain, phylogenetic relationship and 3D structure of *MnHR4* were analyzed by bioinformatics. Quantitative real-time PCR (qRT-PCR) analysis showed that *MnHR4* was highly expressed in the ovary. In different embryo stages, the highest mRNA expression was observed in the cleavage stage (CS). At different individual stages, the mRNA expression of *MnHR4* reached its peak on the fifteenth day after hatching (L15). The in vivo injection of 20-hydroxyecdysone (20E) can effectively promote the expression of the *MnHR4* gene, and the silencing of the *MnHR4* gene increased the content of 20E in *M. nipponense*. The regulatory role of *MnHR4* in 20E synthesis and 20E signaling was further investigated by RNAi. Finally, the function of the *MnHR4* gene in the molting process of *M. nipponense* was studied by counting the molting frequency. After knocking down *MnHR4*, the molting frequency of *M. nipponense* decreased significantly. It was proved that *MnHR4* plays a pivotal role in the molting process of *M. nipponense*.

## 1. Introduction

Endocrine signals play a central role in animal growth and maturation. Although vertebrate growth and development are primarily controlled by thyroid hormones and sex steroids, insect development is controlled by several key hormones and neuropeptides, of which the steroid hormone ecdysteroid is the main regulator [1]. 20-Hydroxyecdysone (20E) is a steroid hormone that was originally discovered in plants and arthropods [2]. The accumulation of research over the past two decades has shown that cholesterol is catalyzed into 20E by the Halloween family of genes (i.e., *Spook*, *Phantom*, *Disembodied*, *Shadow* and *Shade*) [3,4,5]. 20E converts hormonal signals into transcriptional responses through members of the nuclear receptor family, and when 20E binds to the EcR/USP complex, the transcriptional cascade leads to the onset of ecdysis and metamorphosis [6,7]. *Drosophila*, as a classic model organism, has become a model for elucidating the regulatory role of 20E. The 20E pulse directs each molt and metamorphosis of the *Drosophila melanogaster*’s life cycle [8]. Many downstream nuclear receptor transcription factors activated by 20E have been identified in insects [9].

Nuclear receptors contain two common structural elements—a DNA-binding domain (DBD) and a ligand-binding domain (LBD)—which are involved in the regulation of myriad biological processes [10]. In the holometabolous insect *D. melanogaster*, some genes, such as nuclear receptor E75 (*E75*), nuclear receptor E78 (*E78*), hormone receptor 3 (*HR3*), hormone receptor 4 (*HR4*), ecdysis-triggering hormone receptor (*ETHR*) and Fushi tarazu factor-1 (*Ftz-f1*), are transcriptionally regulated by 20E and play a central role in transducing the molting signals [11,12]. The expressions of *HR3* and *E75B* increased with the increase in the 20E titer during *D. melanogaster* pupation, and *βFtz-f1* was activated after the decrease in 20E [13]. In the hemimetabolous insect *Blattella germanica*, the regulatory relationship between nuclear receptor genes *E75*, *HR3*, *HR4* and *Ftz-f1* changed with the fluctuation in the 20E titer [14]. In insects, the function of nuclear receptors activated by 20E has been reported in detail. For example, an *E75A* mutation causes developmental arrest and molting defects in *Drosophila* and the *E75C*-mutant *Drosophila* die in adulthood, whereas *E75B*-mutant individuals survive and reproduce normally [15]. The knockdown of *BgE75* results in *B. germanica* prothoracic gland degeneration and ecdysteroid deficiency [14]. *HR3* is involved in the regulation of chitin synthesis and degradation during *Locusta migratoria* molting [16]. Silencing of *BgFtz-f1* prevents normal molting and development of *B. germanica* [17], and *Ftz-f1* is involved in the regulation of *Leptinotarsa decemlineata*’s pupation by regulating 20E and JH titers [18]. *HR4* plays an important role in *Tribolium castaneum* molting and ovulation [19] and plays a central role in coordinating growth and the maturation of *D. melanogaster* [20]. In addition, the *HR4* gene has been identified in some crustacean transcriptomes. In *Litopenaeus vannamei* and *Daphnia*, *HR4* was identified as an ecdysone signaling response gene [21,22]. In *Callinectes sapidus*, increased ecdysteroid concentration induced *HR4* expression [23]. Altogether, nuclear receptor genes have been extensively studied in insect molting and development, but reports on crustaceans are still lacking.

*Macrobrachium nipponense* (Crustacea, Decapoda) is an important freshwater economic prawn in China [24]. Molting is a pivotal event in the growth of crustaceans [25]. *M. nipponense* grows by molting, but the mechanism of molting is still poorly understood. Therefore, it is of great scientific significance to study the molting mechanism of *M. nipponense* for breeding and increasing production. We previously demonstrated the functions of *Spook* and *Ftz-f1* genes in the molting and ovarian development of *M. nipponense* [5,26]. Transcriptome analysis of the different molting stages of *M. nipponense* revealed that *MnHR4* is an important differential gene. To further improve the molting molecular mechanism of *M. nipponense*, this study investigated whether *MnHR4* functions during the molting process of *M. nipponense*. In the current study, the *MnHR4* gene was identified in *M. nipponense*. The domain, phylogenetic relationship and 3D structure of *MnHR4* were analyzed by bioinformatics. The expression patterns of the *MnHR4* gene in different tissues and developmental stages of *M. nipponense* were detected by qRT-PCR. The expression of the *MnHR4* gene was detected by qRT-PCR after a 20E injection in vivo. After knockdown of the *MnHR4* gene by the RNA interference, the content of 20E in *M. nipponense* was detected by an ELISA. The regulatory role of *MnHR4* in 20E synthesis and 20E signaling was further investigated by RNAi. Finally, the function of the *MnHR4* gene in the molting process of *M. nipponense* was studied by counting the molting frequency.

## 2. Results

### 2.1. Sequence Analysis and Phylogeny of MnHR4

The rapid amplification of the cDNA ends (RACE) (TaKaRa, Kyoto, Japan) of the *HR4* fragment yielded a cDNA sequence with a putative open reading frame of 2901 bp, encoding a total of 966 amino acids, named *MnHR4* (Figure 1). A phylogenetic tree of the *HR4* amino acids of different species was constructed, indicating that insecta and crustacea form two independent clades, which is consistent with the traditional taxonomy of species. *M. nipponense* is the most closely related to *Penaeus chinensis*, followed by *Armadillidium vulgare* (Figure 2). A comparison of the *HR4* amino acid sequences between *M. nipponense* and other crustaceans using DNAMAN 6.0 showed that *MnHR4* contained a conserved C4 zinc finger (ZnF_C4) domain. Zinc finger domains are relatively small protein motifs containing different binding-specific finger-like protrusions, usually in clusters. The *HR4* amino acid identity of *M. nipponense* and *P. chinensis* was 49.64%, followed by *A. vulgare* at 30.18% (Figure 3). The amino acid sequence of *MnHR4* was analyzed using the iterative threading assembly refinement (I-TASSER) server. Figure 4A shows a three-dimensional (3D) atom model of *M. nipponense*’s *MnHR4* generated by I-TASSER. The predicted binding ligands are the pale-green spheres, and the binding residues are the blue ball and stick (Figure 4B).

### 2.2. Expression of the MnHR4 Gene in Different Tissues and Different Developmental Stages

The expression levels of *MnHR4* mRNA in different tissues and different developmental stages of *M. nipponense* were investigated by qRT-PCR. The highest mRNA expression of *MnHR4* was observed in the ovary, followed by the eyestalk, with the lowest expression in the hepatopancreas. The expression levels of *MnHR4* mRNA in the ovary and eyestalk were significantly higher than those in other tissues, and the expression level in the ovary was 4.47-fold higher than that in the eyestalk (*p* < 0.05) (Figure 5A). The expression of *MnHR4* mRNA showed no significant difference in different ovary stages (*p* > 0.05) (Figure 5B). In different embryo stages, the highest mRNA expression was observed in the cleavage stage (CS), followed by the blastula stage (BS), and the lowest was observed in the zoea stage (ZS). The expression level in the CS was 4.53-fold that in the BS and 23.22-fold that in the ZS (*p* < 0.05) (Figure 5C). In different individual stages, the mRNA expression of *MnHR4* gradually decreased from the first day after hatching (L1) to L10 and reached a peak at L15. *MnHR4* mRNA expression was lowest on the first day post-larvae (PL1) and showed a significant difference (*p* < 0.05) (Figure 5D).

### 2.3. Interaction between 20E and MnHR4

Referring to previous studies, 20E (5 μg/g) was injected into *M. nipponense* [26]. The effect of 20E on the expression of *MnHR4* was detected using qRT-PCR (Figure 6A). The results show that there was no significant difference in the expression level of the *MnHR4* gene between the experimental group and the control group at 0 h (no injection) and 3 h after injection (*p* > 0.05). The expression level of the *MnHR4* gene in the experimental group was significantly higher than that in the control group at 6 and 12 h after injection, and it reached a peak at 12 h in the experimental group (*p* < 0.05). There was no significant difference in the expression of *MnHR4* between the two groups at the 24^th^ hour after injection (*p* > 0.05). After knockdown of *MnHR4*, the content of 20E in *M. nipponense* was measured by an ELISA (Figure 6B). The results showed that there was no significant difference in the content of 20E in *M. nipponense* on the first day (*p* > 0.05). The content of 20E in the experimental group of *M. nipponense* was significantly higher than that in the control group on the 5th day. Compared with the control group, the content of 20E in the experimental group increased by 37.79% (*p* < 0.05).

### 2.4. Effects of RNAi MnHR4 on the Expression of Mn-Spook, Phantom, HR3, E75b, ETHR and Mnftz-f1

The regulatory role of *MnHR4* in 20E synthesis and 20E signaling was further investigated by RNAi. After knockdown of *MnHR4*, the expression levels of genes catalyzing 20E synthesis (i.e., *Mn-Spook* and *Phantom*) and downstream genes conducting 20E signaling (i.e., *HR3*, *E75b*, *ETHR* and *MnFtz-f1*) were detected by qRT-PCR (Figure 7). Compared to the control, the expression of *MnHR4* mRNA in the experimental group decreased by 17.45%, 69.25%, 73.88% and 69.97% at 24, 48, 96 and 120 h after ds*MnHR4* administration, respectively (*p* < 0.05) (Figure 7A). After the knockdown of *MnHR4*, the expression levels of *Mn-Spook* and *Phantom* in the experimental group significantly increased. At the 120th hour after the knockdown of *MnHR4*, the expression levels of *Mn-Spook* and *Phantom* in the experimental group were 24.4-fold and 2.1-fold higher than those in the control group, respectively (Figure 7B,C). The results show that the expression levels of *HR3* and *E75b* in the experimental group also significantly increased at the 24th and 48th hours after *MnHR4* gene silencing. At the 120th hour after silencing, the expression of *HR3* in the experimental group was 20.55-fold that of the control group, and the expression of *E75b* in the experimental group was 2.49-fold that of the control group (Figure 7D,E). On the contrary, the expressions of *ETHR* and *MnFtz-f1* in the experimental group decreased to different degrees compared with the control group after *MnHR4* gene silencing. The expression of *ETHR* in the experimental group decreased by 31.13% and 38.91% at the 96th and 120th hour of *MnHR4* gene silencing, respectively (Figure 7F). Compared with the control group, the expression of *MnFtz-f1* in the experimental group decreased by 79.36% at the 120th hour (Figure 7G).

### 2.5. Effect of MnHR4 Knockdown on the Molting Frequency of M. nipponense

Figure 8 shows the molting frequency of *M. nipponense* in the control and experimental groups after *MnHR4* knockdown. *M. nipponense* starts molting on the second day and completes one round of molting on the 12th day. The results show that there was no significant difference in the frequency of molting between the experimental and control groups during the first round of molting (*p* > 0.05). From the 21st day, the control group of *M. nipponense* began the second round of concentrated molting, which was significantly higher than the experimental group of *M. nipponense’s* molting frequency (*p* < 0.05).

## 3. Discussion

Nuclear receptor genes function in many biological processes such as embryonic development, sex determination, insect metamorphosis and molting [29,30,31]. Using RNAi, we previously demonstrated that *MnFtz-f1* played a pivotal role in the molting and ovulation process of *M. nipponense* [26]. Previous studies have demonstrated that *HR4* plays an important role in both holometabolous and hemimetabolous insects [11,14], but its function in *M. nipponense* is still unclear.

In the present study, we identified the nuclear receptor gene *MnHR4* from the transcriptome of *M. nipponense* at different molting stages. The predicted *MnHR4* coding region encodes a total of 966 amino acids, including two conserved domains: the c4 zinc finger in nuclear hormone receptors (ZnF_C4) and ligand-binding domain of hormone receptors (HOLI). Zinc finger domains are relatively small protein motifs with a stable structure and are involved in a wide range of physiological functions including controlling embryonic development and cell differentiation [32,33]. HOLI is located in the LBD region of the nuclear receptor gene, which acts as a molecular switch to turn on transcriptional activity by binding to ligands [34]. In the phylogenetic tree, there is a clear boundary between crustaceans and insects, indicating that *HR4* is more conserved among its class. 

*MnHR4* was expressed in multiple tissues of *M. nipponense*, with higher expression levels in the ovary, indicating that *MnHR4* has different functions in *M. nipponense*. In insects, *HR4* expression was also detected in multiple tissues [35]. The extremely high expression of *HR4* in the ovaries is similar to other studies on *Drosophila*. Recent studies on *Drosophila* have shown that *HR4* is strongly expressed in the ovaries and is required for *Drosophila* to oogenesis [36]. There was no significant difference in the *MnHR4* in different ovarian development stages of *M. nipponense*. Therefore, we speculate that *MnHR4* may be required to function in the whole ovarian development cycle. The expression of *MnHR4* in the cleavage stage was significantly higher than that in other stages of embryonic development, suggesting it has a function during cell division. This result is similar to the expression trend of nuclear receptor gene *MnFtz-f1* in *M. nipponense* [26]. Additionally, *MnHR4* expression levels were the highest at L15 during the larval developmental stages, suggesting that *MnHR4* may function during the metamorphosis of *M. nipponense* [37].

To explore the relationship between 20E and *MnHR4*, the expression level of *MnHR4* was detected after 20E injection in *M. nipponense*, and the content of 20E in *M. nipponense* was measured after *MnHR4* knockdown. The expression of *MnHR4* significantly increased after 20E injection in vivo, which proves that 20E can induce the expression of *MnHR4* in *M. nipponense*. King-jones et al. demonstrated that feeding 20E for 3–4 h induced the peak expression of *HR4* in *Drosophila* [20]. Similarly, the injection of 20E upregulated *HR4*’s expression in *B. germanica* and *L. decemlineata* [38,39]. We further investigated the effect of the knockdown of *MnHR4* on the 20E content. The results show that the knockdown of *MnHR4* increased the 20E titer in *M. nipponense*. In *L. decemlineata*, 20E titers also increased after *HR4* knockdown, which is consistent with our conclusions [39]. Conversely, 20E titers decreased after silencing *HR4* in *L. migratoria* [35], whereas 20E titers were not affected by silencing *HR4* in *B. germanica* [38]. In conclusion, *HR4* may have different regulatory effects on 20E in different species, and its molecular mechanism needs to be further investigated.

RNAi is an effective method to explore the regulatory relationship between genes. We further investigated the molecular mechanism of *MnHR4* in 20E synthesis and signaling by RNAi (Figure 9). *Spook* and *Phantom* are Halloween family member genes that function in the 20E biosynthetic pathway [5,40]. To investigate the molecular mechanism of *MnHR4* on 20E synthesis in *M. nipponense*, the expressions of *Spook* and *Phantom* were detected after knockdown of *MnHR4*. The expressions of *Spook* and *Phantom* significantly increased after knockdown of *MnHR4*, indicating that *MnHR4* has an inhibitory effect on *Spook* and *Phantom*. Similar studies have shown that *HR4* inhibits ecdysone synthesis by regulating cytochrome P450 genes [41]. *HR4* is a repressor of early ecdysone-induced regulatory genes [20]. There are many nuclear receptor genes in the 20E signaling pathway, whose main function is to transmit upstream signals [42]. To further investigate the role of *MnHR4* in 20E signaling, the expression of other genes (i.e., *HR3*, *E75b*, *ETHR* and *MnFtz-f1*) was examined after knockdown of *MnHR4*. The results show that the expressions of *HR3* and *E75b* significantly increased after silencing *MnHR4*, indicating that *MnHR4* had an inhibitory effect on *HR3* and *E75b*. In *L. decemlineata*, the expressions of *HR3* and *E75* were also affected by *HR4*, and the knockdown of *HR4* significantly upregulated the expressions of *HR3* and *E75* [39]. In addition, the expressions of *ETHR* and *MnFtz-f1* were decreased after knockdown of *MnHR4*, indicating that *MnHR4* had a promoting effect on them. Previous studies have demonstrated that *ETHR* functions during the molting process of *M. nipponense* [43], but the regulatory relationship between *HR4* and *ETHR* is rarely reported. Our results demonstrate that *MnHR4* positively regulates *ETHR*, which provides a reference for future research. *Ftz-f1* plays a central role in coordinating different molting processes [44]. Consistent with our conclusion, the abundant results demonstrate that *HR4* can induce the expression of *Ftz-f1* [20,38,41]. In conclusion, *MnHR4* plays a regulatory role in 20E synthesis and signaling. We further investigated whether the molting of the *M. nipponense* occurred after the knockdown of *MnHR4*. It was proved that *MnHR4* plays a pivotal role in the molting process of *M. nipponense*. The molting function of *HR4* has been demonstrated in some insects. In *B. Germanica*, interference with *HR4* resulted in molting failure and eventual death [38]. *HR4* is required for pupal molting and adult oogenesis of *Tribolium castaneum* [19]. The knockdown of *MnHR4* increased 20E titers, which significantly inhibited the molting of *M. nipponense*. Our previous study showed that knockdown of *MnFtz-f1* reduced 20E titers, similarly, leading to the failure of *M. nipponense*’s molting [26]. In *B. mori*, increasing or decreasing 20E titers could affect the normal physiological phenomena of the larvae or even lead to death [45]. Precise regulation of 20E titers is also important for the molting process in *D. melanogaster* [1]. This suggests that an appropriate 20E titer is important for regulating successful molting in insects or crustaceans.

In summary, the *MnHR4* gene was identified in *M. nipponense*. The *MnHR4* gene was comprehensively analyzed using bioinformatics, qRT-PCR, RNAi, ELISA, etc. Our results strongly demonstrate that *MnHR4* functions in the molting process of *M. nipponense* by regulating 20E synthesis and 20E signaling. This study further enriched the molecular regulatory mechanism of molting in *M. nipponense* and could be useful for future gene-editing breeding.

## 4. Materials and Methods

### 4.1. Experimental Prawns and Conditions

Experimental prawns (2.15 ± 0.63 g) were obtained from the Dapu experimental base, Freshwater Fisheries Research Center, Chinese Academy of Fishery Sciences. Briefly, the experimental prawns were transferred from Dapu to the laboratory’s constant-temperature water-circulation system and allowed to acclimatize for a week. Tissues (i.e., ovaries, muscles, gills, hepatopancreas, eyestalks and hearts) were harvested, frozen in liquid nitrogen and stored at −80 ℃ for RNA extraction. Samples were also collected at different stages of embryo, individual and ovarian development according to previous criteria [27,28]. All sampling was performed in triplicate (*n* = 6). The prawns in this study were handled according to the guidelines of the Institutional Animal Care and Use Ethics Committee of the Freshwater Fisheries Research Center, Chinese Academy of Fishery Sciences (Wuxi, China).

### 4.2. Nucleotide Sequence and Bioinformatics Analysis of MnHR4

Total RNA was extracted using the RNAiso Plus kit (TaKaRa, Shiga, Japan), as described previously [26]. DNase I (Sangon, Shanghai, China) was used to eliminate possible DNA contamination, and 1.2% agarose gel and a NanoDrop ND2000 (NanoDrop Technologies, Wilmington, DE, USA) were used to detect RNA quality and concentration, respectively, with an A260/A280 ratio of 1.9–2.0.

The *MnHR4* cDNA fragments were screened from the transcriptome of *M. nipponense* at different molting stages. The *MnHR4* cDNA fragments were analyzed using the GenBank BLASTX and BLASTN programs (https://blast.ncbi.nlm.nih.gov/Blast.cgi (accessed on 20 April 2022)). The open reading frame (ORF) of *MnHR4* was predicted by using ORF Finder (https://www.ncbi.nlm.nih.gov/orffinder/ (accessed on 20 April 2022)). Molecular Evolutionary Genetics Analysis (MEGA-X) software was used to construct a phylogenetic tree, and the bootstrapping replications were 1000 [46,47]. DNAMAN 6.0 was used for translating and aligning amino acid sequences. The spatial structure and function of *MnHR4* amino acids were predicted by I-TASSER (https://zhanglab.ccmb.med.umich.edu/I-TASSER/ (accessed on 22 April 2022)) [48]. The *HR4* amino acid sequences of other species investigated in this study were downloaded from the GenBank database (http://www.ncbi.nlm.nih.gov/ (accessed on 20 April 2022)).

### 4.3. Quantitative Real-Time PCR (qRT-PCR) Analysis

Gene expression patterns were evaluated using qRT-PCR on a Bio-Rad iCycler iQ5 Real-Time PCR System (Bio-Rad, Carlsbad, CA, USA). The primers were designed using NCBI’s Primer-Blast tool (http://www.ncbi.nlm.nih.gov/tools/primer-blast/ (accessed on 25 April 2022)) and synthesized by exsyn-bio Technology Co., Ltd. (Shanghai, China). The reaction system and procedure of qRT-PCR have been described in previous studies [5]. The internal reference gene, eukaryotic translation initiation factor 5A (*EIF*), was used as a control for data normalization [49], and the relative expression levels of the genes were calculated with the 2^−ΔΔCT^ method [50].

### 4.4. RNA Interference (RNAi)

The design and synthesis of *MnHR4*-interfering primers followed a method used in previous studies [26]. The green fluorescent protein gene (GFP) was used as a control [51]. A total of 180 healthy female prawns in the pre-molting stage were randomly divided into two groups (i.e., experimental group and control group) in triplicate (*n* = 30). The experimental group and control group were injected with *MnHR4* dsRNA and *dsGFP*, respectively (8 µg/g of body weight). Prawn tissues were collected in triplicate at 24, 48, 96 and 120 h after injection for RNA extraction and an interference efficiency assessment (*n* = 6). On the basis of the significant interference efficiency of the experimental group and control group, 300 prawns in the pre-molting stage were divided into two groups using the same method and injected with *MnHR4* dsRNA and *dsGFP*, respectively (*n* = 50). The number of molting prawns per day was counted, and the molting frequency was calculated. The number of molts was counted for 30 days, and dsRNA was injected every five days. Molting frequency = (Nm/Ns)/D, where Nm is the total number of molts; Ns is the number of prawns in the aquarium; and D is the number of experimental days [5,26].

### 4.5. ELISA

After knocking down the *MnHR4* gene, the Shrimp EH ELISA Kit (lot number: m1963525-J; Meibo, Shanghai, China) was used to detect the content of 20E in *M. nipponense*, according to the manufacturer’s protocols. In brief, the tissue was first washed with pre-cooled PBS to remove the residual blood. Next, the tissue was weighed and cut into pieces, and it was added to the PBS at a ratio of 1:9. Then, it was put into a glass homogenizer and fully ground on ice to lyse the tissue’s cells. The tissue homogenate was centrifuged at 5000 rpm for 5–10 min, and the supernatant was removed until use. All reagents and samples were prepared before the experiment: (1) The standard wells and sample wells were set on the microtiter plate. Fifty microliters of different concentrations of standards (i.e., 2000, 1000, 500, 250, 125 and 62.5 pg/mL) were added to the standard wells, and 50 μL of the samples to be tested were added to the sample wells; blank wells were not added. (2) Then, 100 μL of enzyme ligands was added to the standard and sample wells, and the reaction wells were sealed with a closure plate membrane and incubated for 60 min at 37 ℃ in a water bath or incubator. (3) The microtiter plate was rinsed 4–5 times, and then, 50 μL of substrate A and B was added to each well. They were mixed gently and incubated at 37 °C for 15 min. (4) After adding 50 μL of stop solution to each well and tapping the plate to ensure adequate mixing, the OD values of each well were measured at a wavelength of 450 nm within 15 min. (5) Taking the OD value of the standard as the abscissa and the concentration value of the standard as the ordinate, the standard curve was drawn using Excel software, and the linear regression equation was obtained. The OD value of the sample was substituted into the equation to calculate the concentration of the sample.

### 4.6. 20E Treatments

A total of 120 pre-molting *M. nipponense* were divided into the experimental group and the control group, in triplicate (*n* = 20). *M. nipponense* in the experimental group was injected with 20E (Sigma-Aldrich, St. Louis, MO, USA; 5 μg/g), and the control group was injected with an equal volume of solvent (ethanol) [26]. Prawn tissues were collected at 3, 6, 12, 24 and 48 h after injection of 20E and stored at −80 °C after liquid nitrogen quick-freezing for mRNA extraction. The relative expression levels of the *MnHR4* gene in the experimental and control groups were evaluated by qRT-PCR.

### 4.7. Data Analysis

Statistical analyses were performed using SPSS 20.0 software (IBM, New York, NY, USA). One-way ANOVA was used for comparisons between multiple sample means. The differences between the two groups were compared using the independent sample *t*-test. All data are presented as the mean ± SEM. The significance level for the data was set at *p* < 0.05.

## Figures and Tables

**Figure 1 ijms-23-12528-f001:**
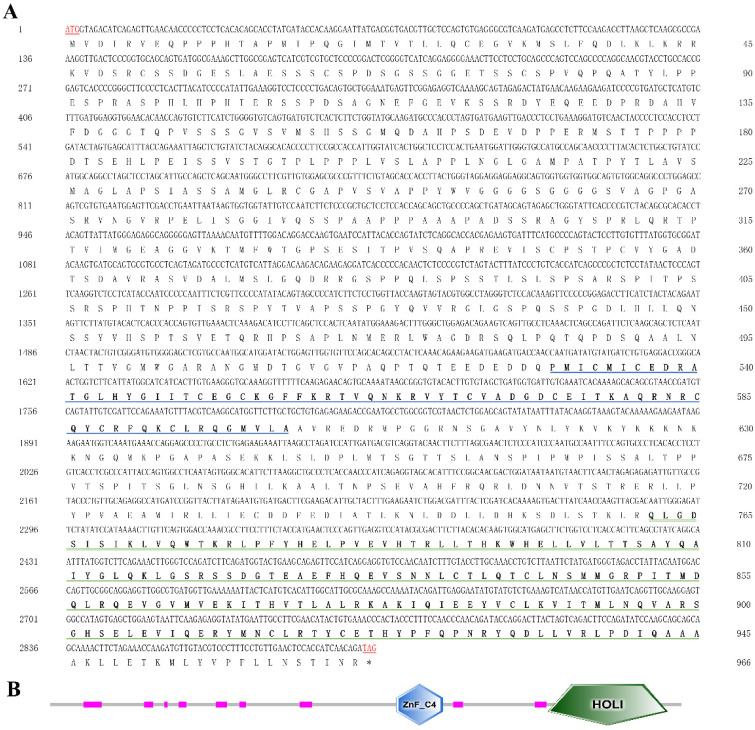
(**A**) Nucleotide and deduced amino acid sequences of *MnHR4* in *M. nipponense*. The numbers on the left and right refer to the coordinates of the nucleotide and amino acid sequences, respectively. Blue and green underscores represent the ZnF_C4 and HOLI domains, respectively. The termination signals are indicated with an asterisk (*). (**B**) Domain architecture organization of *MnHR4* as predicted by SMART (http://smart.embl-heidelberg.de/ (accessed on 20 April 2022)). The conserved domains (i.e., ZnF_C4 and HOLI) of MnHR4 are shown by the different shapes and colors.

**Figure 2 ijms-23-12528-f002:**
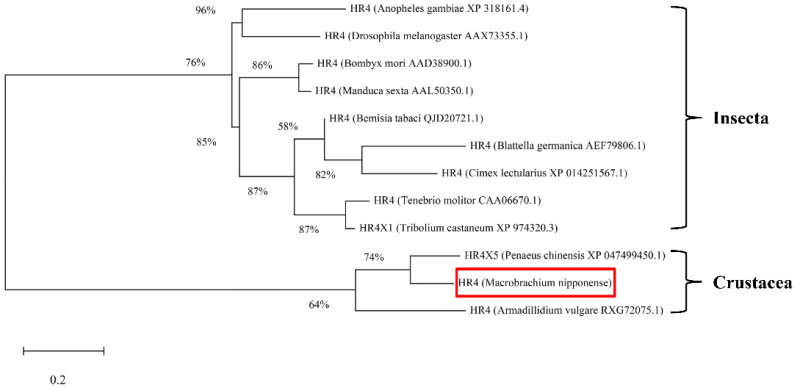
Phylogenetic tree of the amino acid sequences of *HR4* from various species. The numbers shown at the branches indicate the bootstrap values (%). The red rectangles indicate the position of *M. nipponense* in the phylogenetic tree. The species and GenBank accession numbers for constructing the phylogenetic tree are listed below: *Anopheles gambiae* (XP_318161.4), *D. melanogaster* (AAX73355.1), *Bombyx mori* (AAD38900.1), *Manduca sexta* (AAL50350.1), *Bemisia tabaci* (QJD20721.1), *B. germanica* (AEF79806.1), *Cimex lectularius* (XP_014251567.1), Tenebrio molitor (CAA06670.1), *Tribolium castaneum* (XP_974320.3), *P. chinensis* (XP_047499450.1), *M. nipponense* and *A. vulgare* (RXG72075.1).

**Figure 3 ijms-23-12528-f003:**
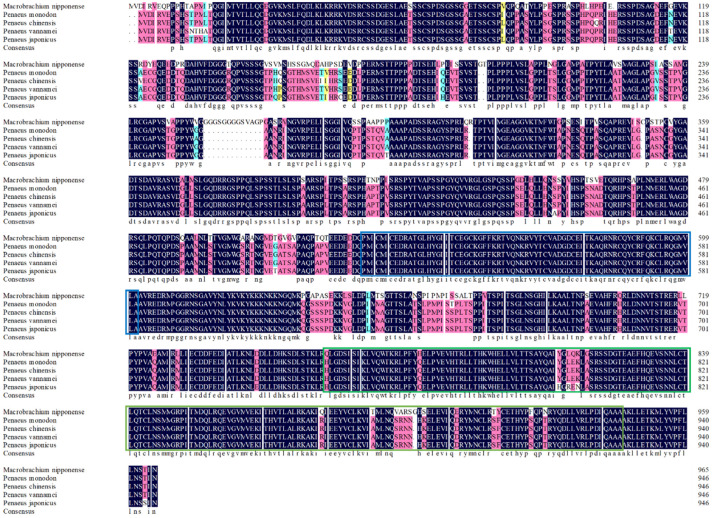
Sequence alignment of the *HR4* amino acids between *M. nipponense* and other crustaceans. The blue and green underscores represent the ZnF_C4 and HOLI domains, respectively.

**Figure 4 ijms-23-12528-f004:**
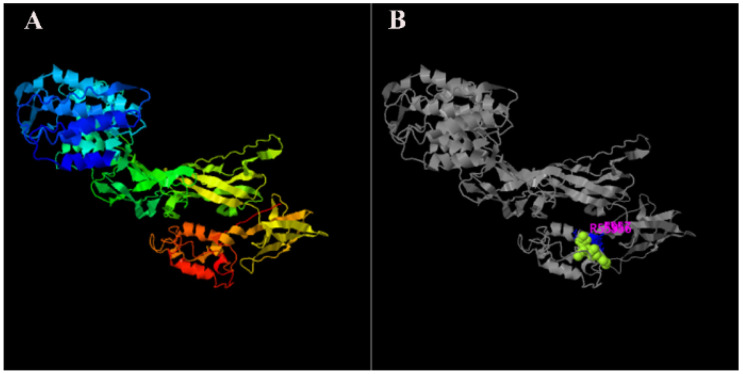
The 3D structures of *MnHR4* predicted by I-TASSER: (**A**) predicted function of *MnHR4* 3D structures using COFACTOR and COACH, where the pale-green sphere represents the predicted binding ligand; (**B**) molecules and ions that bind to anchoring proteins are called ligands.

**Figure 5 ijms-23-12528-f005:**
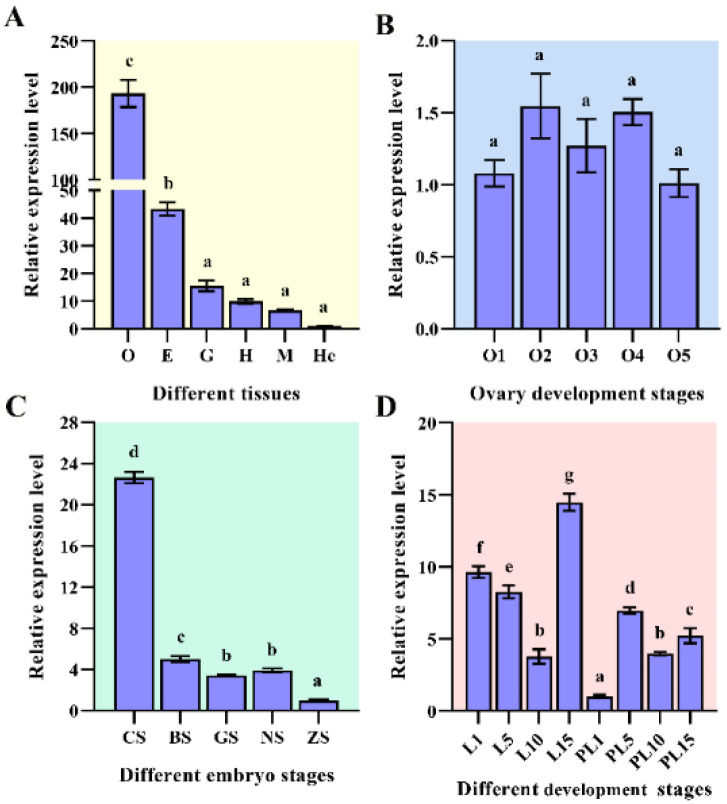
Expression of *MnHR4* mRNA in different tissues and different developmental stages of *M. nipponense*. Samples (at different developmental stages) were collected at the experimental site in Dapu, according to previous criteria [27,28]. The expression of *MnHR4* mRNA was normalized to the *EIF* transcript level. (**A**) Different tissues: O, ovary; E, eyestalk; G, gill; H, heart; M, muscle; He, hepatopancreas. (**B**) Different ovary stages: O1, undeveloped stage; O2, developing stage; O3, nearly ripe stage; O4, ripe stage; O5, spent stage. (**C**) Different embryo stages: CS, cleavage stage; BS, blastula stage; GS, gastrula stage; NS, nauplius stage; ZS, zoea stage. (**D**) Different development stages: L1, the first day after hatching; PL1, the first day post-larvae, etc. Statistical analyses were performed by one-way ANOVA. Data are shown as the mean ± SEM (*n* = 6). Bars with different letters indicate significant differences (*p* < 0.05).

**Figure 6 ijms-23-12528-f006:**
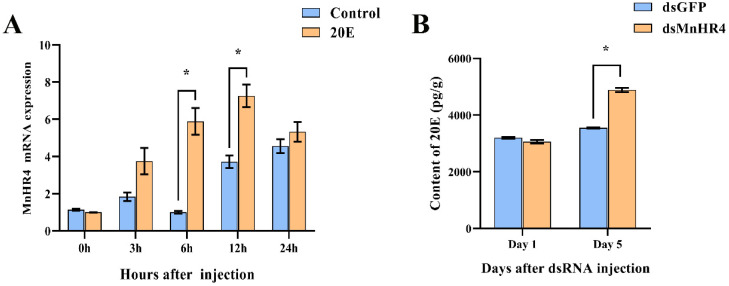
(**A**) Expression of *MnHR4* mRNA in the ovary under the influence of 20E. The expression of *MnHR4* mRNA was normalized to the *EIF* transcript level. (**B**) The content of 20E in *M. nipponense* after knockdown of *MnHR4*. Control, injection of solvent; 20E, injection of 20E. Data are expressed as the mean ± SEM (*n* = 6). Significant differences between the experimental group and the control group were determined using Student’s *t*-test (* *p* < 0.05).

**Figure 7 ijms-23-12528-f007:**
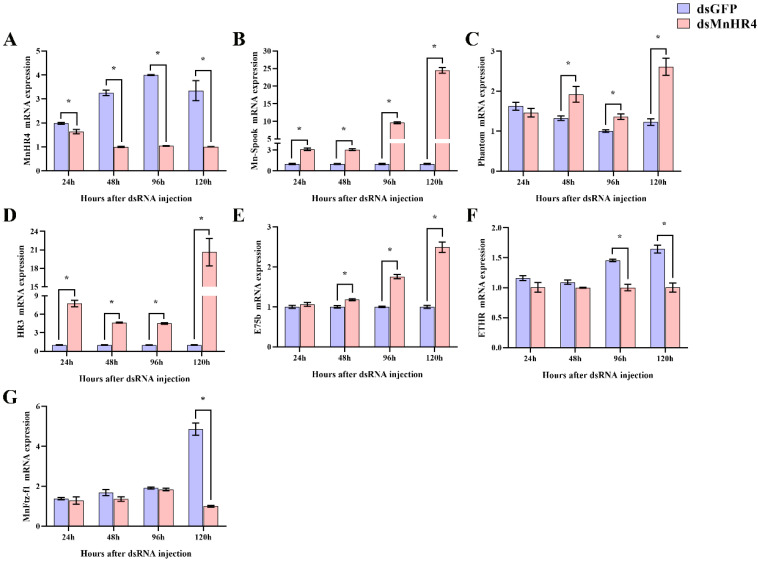
Knockdown of *MnHR4* on the expression of other 20E-related genes in the ovary of *M. nipponense*. The expression of *MnHR4* mRNA was normalized to the *EIF* transcript level: (**A**) *MnHR4*; (**B**) *Mn-Spook*; (**C**) *Phantom*; (**D**) *HR3*; (**E**) *E75b*; (**F**) *ETHR*; (**G**) *Mn-Ftz-f1*. Data are expressed as the mean ± SEM (*n* = 6). Significant differences between the experimental group and the control group were determined using Student’s *t*-test (* *p* < 0.05).

**Figure 8 ijms-23-12528-f008:**
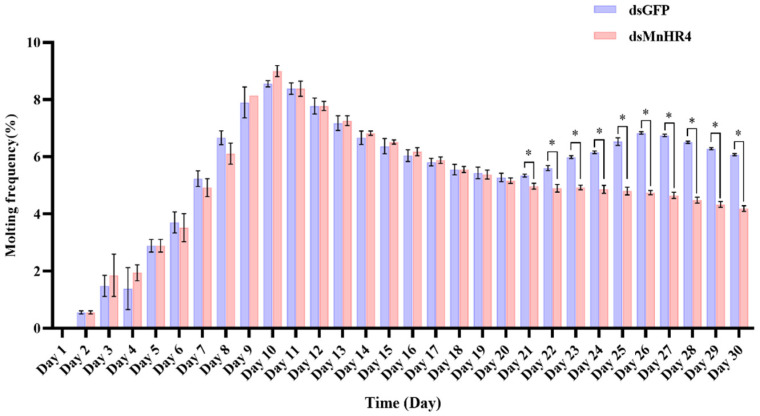
Molting frequency of *M. nipponense* in the experimental and control groups after knocking down *MnHR4*. Molting frequency = (Nm/Ns)/D, where Nm is the total number of molts; Ns is the number of prawns in the aquarium; D is the number of experimental days. Data are expressed as the mean ± SEM. Significant differences between the experimental group and the control group were determined using Student’s *t*-test (* *p* < 0.05).

**Figure 9 ijms-23-12528-f009:**
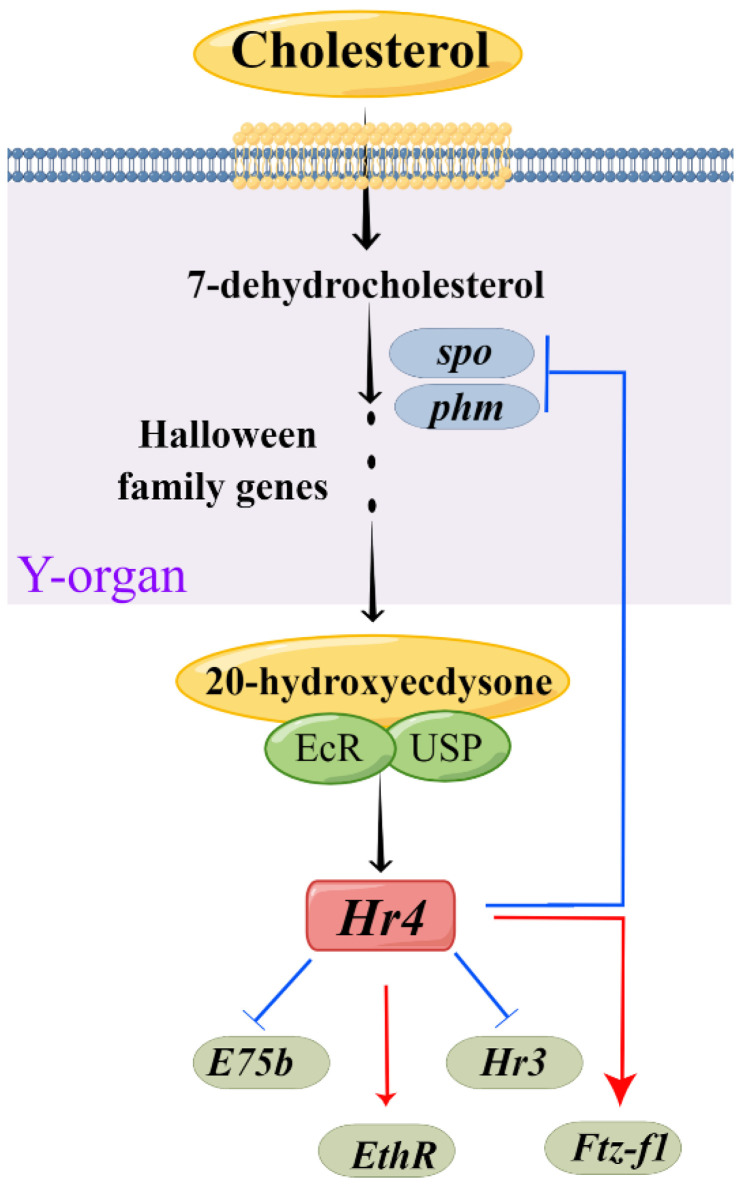
Model for *MnHR4* function, developed using Figdraw (www.figdraw.com). The Y-organ is a pair of secretory glands of crustaceans, the site of the synthesis of cholesterol into ecdysone.

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
