# Peer review of "MnHR4 Functions during Molting of Macrobrachium nipponense by Regulating 20E Synthesis and Mediating 20E Signaling"

_ijms, 2022, doi:10.3390/ijms232012528_

Round 1

Reviewer 1 Report

     Ecdysteroids play essential roles in arthropod physiology, especially in molting, metamorphosis, and oogenesis. There biosynthesis and signaling pathways were deeply studied in insects, but they are poorly understood in crustaceans. Authors identified HR4gene (MnHR4), which is important in ecdysteroid signaling, in a freshwater prawn, Macrobrachium nipponense. The expression patterns of MnHR4suggested that it has important functions in embryonic/post-embryonic development and oogenesis. Injection of 20E, the active form of ecdysteroid, induced MnHR4expression, and knockdown of MnHR4affected expressions of several ecdysteroid-signaling factors. These results showed that MnHR4 functions as a mediator of ecdysteroid signaling. It was also shown that MnHR4knockdown induced ecdysteroid-biosynthetic genes and 20E titer, indicating MnHR4function in ecdysteroid biosynthesis. The effect of MnHR4 knockdown on molting frequency was also studied. The current study is valuable for publication, although this reviewer feels several concerns in the manuscript as follows:

1. Nomenclature

Some genes including HR4are called with “Mn” from the species name, but others are not. This is not consistent. 

2. Effect of MnHR4 knockdown on 20E biosynthesis and molting

The knockdown of MnHR4 induced the expression of spo/phmand 20E titer. However, MnHR4knockdown inhibited individual molting (Fig. 8). This seems inconsistent. The explanation and discussion of these results are required. 

3. ETHR

Although ETHR was studied as a nuclear receptor, it is NOT a nuclear receptor at least in insects. This should be fixed. 

4. real-time PCR

In the DeltaDeltaCt method, reference samples are required and the relative expression is shown as relative value to the reference sample. In the manuscript, reference samples never indicated. Readers can not understand what expression value = 1 means. For examples it is unclear whether the Y-axes in Fig. 5 are comparable to each other. 

Other minor concerns are listed below:

1. Primer sequences

Primer sequences should be shown. They can be supplementary information available on line.

2. Page 1, line 16

The domain, phylogenetic relationship, and 3D-structure seem to be analyzed for MnHR4 protein. In the manuscript, it is said they are analyzed for the gene. The usage of the term “gene” is also inappropriate in other sentences. 

3. Page 2, line 5

This Drosophilamust be Drosophila melanogaster. There are many Drosophilain the manuscript. If they are not descriptions generally for genus Drosophila, but for a species D. melanogaster, they should be corrected. 

4. Page 3, line 101

The source of the transcriptome data should be shown. 

5. Page 3, line 122

If possible, the developmental timing of individual samples connected with molt timing is desired. Please describe that they were just after a molt, intermolt stage, before a molt, or it was not controlled. 

6. Page 3, line 138

20. E should be 20E. 

7. Page 4, line 153

It should be shown which RACE system was used. 

8. Page 4, line 167

Is the “4B” “Figure 4B”?

9. Figure 1

There is no description about Fig. 1B in the Fig. 1 legend. 

10. Figure 6/7

The tissues used for expression analyses should be described.

11. Figure 8

“Molting frequency” is not clear. Is this the population of animals that molt on each day? If so, “population of molting animals” or another expression might be better. It is also important to explain it in the legend. 

12. Page 9, line 273

The use of “novel” is not appropriate, because HR4 is a well-known factor. 

13. Page 10, line 304/306

L. Decemlineata” and “L. Migratoria” should be “L. decemlineata” and “L. migratoria”.

14. Figure 9

It is not clear what the inhibitory line from Hr4 to 20E means. In this study, inhibition of spo/phmby HR4 was shown, but other inhibitory systems were not studied. Spo/Phm should not be capitalized if they are gene names. Actually, Phm starting with capital P indicates another gene, Peptidylglycine-α-hydroxylating monooxygenase in D. melanogaster. “Y-orgen” is “Y-organ”, and this organ should be explained briefly in the legend or the introduction, because not all the readers are crustacean physiologists. 
